# Continuous Surface Embeddings

**Natalia Neverova**    **David Novotny**    **Vasil Khalidov**    **Marc Szafraniec**
**Patrick Labatut**    **Andrea Vedaldi**
{nneverova, dnovotny, vkhalidov, mszafraniec, plabatut, vedaldi} fb.com
Facebook AI Research

## Abstract

In this work, we focus on the task of learning and *representing dense correspondences* in deformable object categories. While this problem has been considered before, solutions so far have been rather ad-hoc for specific object types (i.e., humans), often with significant manual work involved. However, scaling the geometry understanding to all objects in nature requires more automated approaches that can also express correspondences between related, but geometrically different objects. To this end, we propose a new, learnable image-based representation of dense correspondences. Our model predicts, for each pixel in a 2D image, an embedding vector of the corresponding vertex in the object mesh, therefore establishing dense correspondences between image pixels and 3D object geometry. We demonstrate that the proposed approach performs on par or better than the state-of-the-art methods for dense pose estimation for humans, while being conceptually simpler. We also collect a new in-the-wild dataset of dense correspondences for animal classes and demonstrate that our framework scales naturally to the new deformable object categories.

## 1  Introduction

Understanding the geometry of natural objects, such as humans and other animals, must start from the notion of *correspondence*. Correspondences tell us which parts of different objects are geometrically equivalent, and thus form the basis on which an understanding of geometry can be developed. In this paper, we are interested in particular in learning and computing correspondence starting from 2D images of the objects, a preliminary step for 3D reconstruction and other applications.

While the correspondence problem has been considered many times before, most solutions still involve a significant amount of manual work. Consider for example a state-of-the-art method such as DensePose [17]. Given a new object category to model with DensePose, one must start by defining a *canonical shape S*, a sort of 'average' 3D shape used as a reference to express correspondences. Then, a dataset of images of the object must be collected and annotated with millions of manual point correspondences between the images and the canonical 3D model. Finally, the model must be manually partitioned into a number of parts, or charts, and a deep neural network must be trained to segment the image and regress the $uv$ coordinates for each chart, guided by the manual annotations, yielding a DensePose predictor. Given a new category, this process must be repeated from scratch.

There are some obvious scalability issues with this approach. The most significant one is that the entire process must be repeated for each new object category one wishes to model. This includes the laborious step of collecting annotations for the new class. However, categories such as animals share significant similarities between them; for instance, recently [44] has shown that DensePose trained on humans transfers well on chimpanzees. Thus, a much better scalable solution can be obtained by *sharing* training data and models between classes. This brings us to the second shortcoming of DensePose: the nature of the model makes it difficult to realize this information sharing. In par-

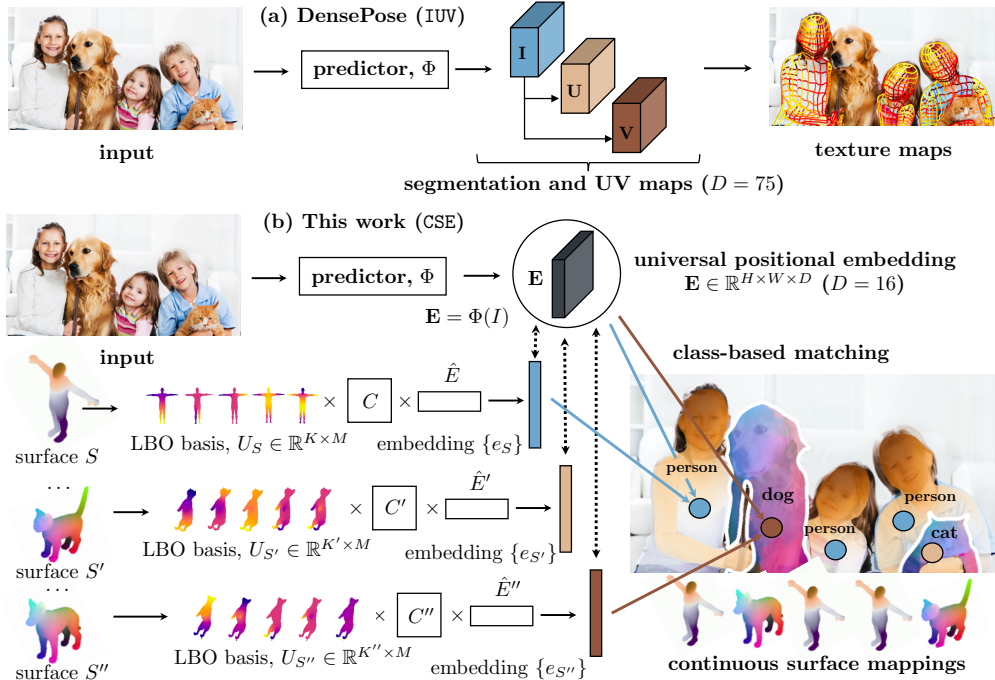

Figure 1: **Overview.** Compared to DensePose [17], our CSE framework is conceptually simpler and is directly extendable to multi-class problems through learning a universal dense pose network. (Color coding for each class mesh in the figure is chosen arbitrarily and independently from others.)

ticular, the need for breaking the canonical 3D models into different charts makes relating different models cumbersome, particularly in a learning setup.

One important contribution of this paper is to introduce a better and more flexible representation of correspondences that can be used as a drop-in replacement in architectures such as DensePose. The idea is to introduce a *learnable positional embedding*. Namely, we associate each point $X$ of the canonical model $S$ to a compact *embedding vector* $e_X$, which provides a deformation-invariant representation of the point identity. We also note that the embedding can be interpreted as a smoothly-varying function defined over the 3D model $S$, interpreted as a manifold. As such, this allows us to use the machinery of *functional maps* [39] to work with the embeddings, with two important advantages: (1) being able to significantly reduce the dimensionality of the representation and (2) being able to efficiently relate representations between models of different object categories.

Empirically, we show that we can learn a deep neural network that predicts, for each pixel in a 2D image, the embedding vector of the corresponding object point, therefore establishing dense correspondences between the image pixels and the object geometry. For humans, we show that the resulting correspondences are as or more accurate than the reference state-of-the-art DensePose implementation, while achieving a significant simplification of the DensePose framework by removing the need of charting the model. As an additional bonus, this removes the 'seams' between the parts that affect DensePose. Then, we use the ability of the functional maps to relate different 3D shapes to help transferring information between different object categories. With this, and a very small amount of manual training data, we demonstrate for the first time that a *single (universal) Dense-Pose network* can be extended to capturing multiple animal classes with a high degree of sharing in compute and statistics. The overview of our method with learning *continuous surface embeddings* (CSE) is shown in Fig. 1b (for comparison, the DensePose [17] setup (IUV) is shown in Fig. 1a).

## 2 Related work

**Human pose recognition.** With deep learning, image-based human pose estimation has made substantial progress [52, 37, 12], also due to the availability of large datasets such as COCO [31],

MPII [3], Leeds Sports Pose Dataset (LSP) [23, 24], PennAction [58], or PoseTrack [2]. Our work is most related with DensePose [17], which introduced a method to establish dense correspondences between image pixels and points on the surface of the average SMPL human mesh model [32].

**Unsupervised pose recognition.** Most pose estimators [5, 49, 7, 50, 43, 48, 33, 59, 22] require full supervision, which is expensive to collect, especially for a model such as DensePose. A handful of works have tackled this issue by seeking unsupervised and weakly-supervised objectives, using cues such as equivariance to synthetic image transformations. The most relevant to us is Slim Dense-Pose [36], which showed that DensePose annotations can be significantly reduced without incurring a large performance penalty, but did not address the issue of scaling to multiple classes.

**Animal pose recognition.** Compared to humans, animal pose estimation is significantly less explored. Some works specialise on certain animals (tigers [30], cheetahs [35] or drosophila melanogaster flies [19]). Tulsiani et al. [51] transfer pose between annotated animals and un-annotated ones that are visually similar. Several works have focused on animal landmark detection. Rashid et al. [41] and Yang et al. [55] studied animal facial keypoints. A well explored class are birds due to the CUB dataset [53]. Some works [57, 45] proposed various types of detectors for birds, while others explored reconstructing sparse [38] and dense [26, 25, 13] 3D shapes. Beyond birds, Zuffi et al. [61, 62, 60] have explored systematically the problem of reconstructing 3D deformable animal models from image data. They utilise the SMAL animal shape model, which is an analogue for animals of the more popular human SMPL model for humans [32]. While [61, 62] are based on fitting 2D keypoints at test time, Biggs et al. [6] directly regresses the 3D shape parameters instead. Recently, Kulkarni et al. [28, 29] leveraged canonical maps to perform the 3D shape fitting as well as establishing of dense correspondences across different instances of an animal species.

**3D shape analysis.** Our work is also related to the literature that studies the intrinsic geometry of 3D shapes. Early approaches [15, 9] analysed shapes by performing multi-dimensional scaling of the geodesic distances over the shape surface, as these are invariant to isometric deformations. Later, Coifman and Lafon [14] popularized the diffusion geometry due to its increased robustness to small perturbations of the shape topology. The seminal work of Rustamov [42] proposed to use the eigenfunctions of the Laplace-Beltrami operator (LBO) on a mesh to define a basis of functions that smoothly vary along the mesh surface. The LBO basis was later leveraged in other diffusion descriptors such as the heat kernel signature (HKS) [47], wave kernel signature (WKS) [4] or Gromov-Hausdorff descriptors [10]. A scale-invariant version of HKS was introduced in [11], while [8] proposed an HKS-based equivalent of the image BoW descriptor [46]. While HKS/WKS establish 'hard' correspondences between individual points on shapes, Ovsjanikov et al. [39] introduced the functional maps (FM) that align shapes in a soft manner by finding a linear map between spaces of functions on meshes. Interestingly, [39] has revealed an intriguing connection between FMs and their efficient representation using the LBO basis. The FM framework became popular and was later extended in [40, 27, 1]. Relevantly to us, [34] proposed ZoomOut, a method that estimates FM in a multi-scale fashion, which we improve for our species-to-species mesh correspondences.

## 3 Method

We propose a new approach for representing continuous correspondences (or surface embeddings, CSE) between an image and points in a 3D object. To this end, let $S \subset \mathbb{R}^3$ be a *canonical surface*. Each point $X \in S$ should be thought of as a 'landmark', i.e. an object point that can be identified consistently despite changes in viewpoint, object deformations (e.g. a human moving), or even swapping an instance of the object with another (e.g. two different humans).

In order to express these correspondences, we consider an **embedding function** $e : S \to \mathbb{R}^D$ associating each 3D point $X \in S$ to the corresponding $D$-dimensional vector $e_X$. Then, for each pixel $x \in \Omega$ of image $I$, we task a deep network $\Phi$ with computing the corresponding embedding vector $\Phi_x(I) \in \mathbb{R}^D$. From this, we recover the corresponding canonical 3D point $X \in S$ probabilistically, via a softmax-like function:

$$p(X|x, I, e, \Phi) = \frac{\exp\left(-\langle e_X, \Phi_x(I)\rangle\right)}{\int_S \exp\left(-\langle e_X, \Phi_x(I)\rangle\right) dX}. \tag{1}$$

In this formulation, the embedding function $e_X$ is learnable just like the network $\Phi$. The simplest way of implementing this idea is to approximate the surface $S$ with a mesh with vertices

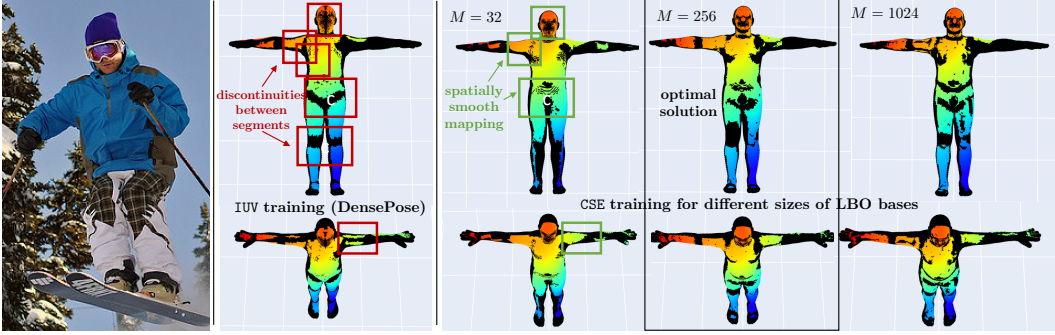

Figure 2: **Comparison of** `IUV` **(DensePose** [17]**), and** `CSE`**-learned mappings** (vertices predicted in the image are shown in color). `CSE` training produces smooth seamless predictions with no shrinking effect (the remaining discontinuities are due to biases in annotations, see the suppl. mat. for details).

$X_1, \ldots, X_K \in S$, obtaining a discrete variant of this model:

$$p(k|x, I, E, \Phi) = \frac{\exp\left(-\langle e_k, \Phi_x(I) \rangle\right)}{\sum_{k=1}^{K} \exp\left(-\langle e_k, \Phi_x(I) \rangle\right)}, \tag{2}$$

where $e_k = e_{X_k}$ is a shorthand notation of the embedding vector associated to the $k$-th vertex of the mesh and $E$ is the $K \times D$ matrix with all the embedding vectors $\{e_k\}_{k=1}^{K}$ as rows.

Given a training set of triplets $(I, x, k)$, where $I$ is an image, $x$ a pixel, and $k$ the index of the corresponding mesh vertex, we can **learn** this model by minimizing the cross-entropy loss:

$$\mathcal{L}(E, \Phi) = - \underset{(I,x,k)\in\mathcal{T}}{\text{avg}} \log p(k|x, I, E, \Phi). \tag{3}$$

We found it beneficial to modify this loss to account for the geometry of the problem, minimizing the cross entropy between a 'Gaussian-like' distribution centered on the ground-truth point $k$ and the predicted posterior:

$$\mathcal{L}_\sigma = - \underset{(I,x,k)\in\mathcal{T}}{\text{avg}} \sum_{q=1}^{K} g_S(q; k) \log p(q|x, I, E, \Phi), \quad g_S(q; k) \propto \exp\left(-\frac{1}{2\sigma^2} d_S(X_q, X_k)\right) \tag{4}$$

where $d_S : S \times S \to \mathbb{R}_+$ is the *geodesic distance* between points on the surface.

**Comparison with DensePose.** DensePose [17], which is the current state-of-the-art approach for recovering dense correspondences, decomposes the canonical model $S = \cup_{i=1}^{J} \tilde{S}_i$ into $J$ non-overlapping patches each parametrized via a chart $f_i : \tilde{S}_i \to [0,1]^2$ mapping the patch $\tilde{S}_i$ to the square $[0,1]^2$ of local $uv$ coordinates. DensePose then learns a network that maps each pixel $x$ in an image $I$ of the object to the corresponding chart index and $uv$ coordinate as $\Phi_x(I) = (i, u, v) \in \{1, \ldots, M\} \times [0,1]^2$.

An issue with this formulation is that the charting $(\tilde{S}_i, f_i)$ of the canonical model $S$ is arbitrary and needs to be defined manually. The DensePose network $\Phi$ then needs to output, for each pixel $u$, a probability value that the pixel belongs to one of the $J$ charts, plus possible chart coordinates $(u, v) \in [0,1]^2$ for each patch. This requires at minimum $J + 2$ values for each pixel, but in practice is implemented as a set of $3J$ channels as different patches use different coordinate predictors. Our model eq. (2) is a substantial simplification because it only requires the initial mesh, whereas the embedding matrix $E$ is learned automatically. This reduces the amount of manual work required to instantiate DensePose, removes the seams between the different charts $\tilde{S}_i$, that are arbitrary, and, perhaps most importantly, allows to share a single embedding space, and thus network, between several classes, as discussed below.

## 3.1 Injecting geometric knowledge via spectral analysis

There are three issues with the formulation we have given so far. First, the embedding matrix $E$ is large, as it contains $D$ parameters for each of the $K$ mesh vertices. Second, the representation

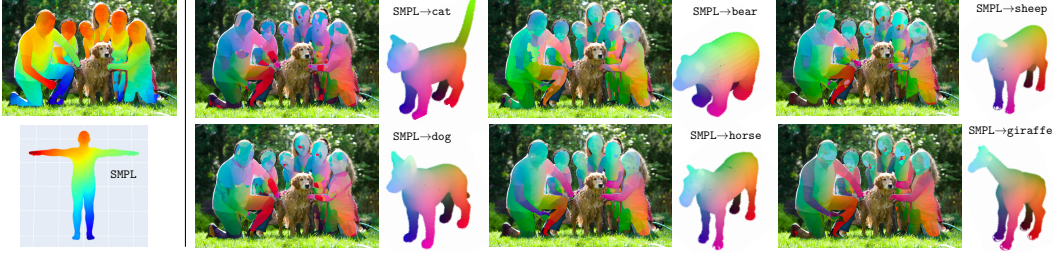

Figure 3: **Initializing animal predictor from a human-trained network and mesh mappings** by setting $\hat{E}' = C\hat{E}$ for each animal class (see 3.3 for details). For each image-mesh pair, the color coding corresponds to the new target animal category, as displayed on the corresponding 3D model.

depends on the discretization of the mesh, so for example it is not clear what to do if we resample the mesh to increase its resolution. Third, it is not obvious, given two surfaces $S$ and $S'$ for two different object categories (e.g. humans and chimps), how their embeddings $e$ and $e'$ can be related.

We can elegantly solve these issues by interpreting $e_X$ as a smooth function $S \to \mathbb{R}^D$ defined on a manifold $\mathcal{S}$, of which the mesh is a discretization. Then, we can use the machinery of *functional maps* [39] to manipulate the encodings, with several advantages.

For this, we begin by discretizing the representation; namely, given a real function $r : S \to \mathbb{R}$ defined on the surface $S$, we represent it as a $K$-dimensional vector $\mathbf{r} \in \mathbb{R}^K$ of samples $r_k = r(X_k)$ taken at the vertices. Then, we define the *discrete Laplace-Beltrami operator* (LBO) $L = A^{-1}W$ where $A \in \mathbb{R}^{K \times K}_+$ is diagonal and $W \in \mathbb{R}^{K \times K}$ positive semi-definite (see below for details). The interest in the LBO is that it provides a basis for analyzing functions defined on the surface. Just like the Fourier basis in $\mathbb{R}^n$ is given by the eigenfunctions of the Laplacian, we can define a 'Fourier basis' on the surface $S$ as the matrix $U \in \mathbb{R}^{K \times K}$ of generalized eigenvectors $WU = AU\Lambda$, where $\Lambda$ is a diagonal matrix of eigenvalues. Due to $A$, the matrix $U$ is orthonormal w.r.t. the metric $A$, in the sense that $U^\top AU = I$. Each column $u_k$ of the matrix $U$ is a $K$-dimensional vector defining an eigenfunction on the surface $S$, corresponding to eigenvalue $\lambda_k \geq 0$. The 'Fourier transform' of function $\mathbf{r}$ is then the vector of coefficients $\hat{\mathbf{r}}$ such that $\mathbf{r} = U\hat{\mathbf{r}}$. Furthermore, if we retain in $U$ only the columns corresponding to the $M$ smallest eigenvalues, so that $U \in \mathbb{R}^{K \times M}$, setting $\mathbf{r} = U\hat{\mathbf{r}}$ defines a smooth (low-pass) function. Appendix A.1 explains how we construct $W$ and $A$ in detail.

This machinery is of interest to us because we can regard the $i$-th component $(e_k)_i$ of the embedding vectors as a scalar function defined on the mesh. Furthermore, we expect this function to *vary smoothly*, meaning that we can express the overall code matrix as $E = U\hat{E}$, where $\hat{E} \in \mathbb{R}^{M \times D}$ is a matrix of coefficients *much* smaller than $E \in \mathbb{R}^{K \times D}$. In other words, we have used the geometry of the mesh to dramatically reduce the number of parameters to learn. LBO bases can also be used to relate different meshes, as explained next.

## 3.2 Relating different categories

Let $S$ and $S'$ be the canonical shapes of two objects, e.g. human and chimp. The shapes are analogous, but also sufficiently different to warrant modelling them with related but distinct canonical shapes. In the discrete setting, a *functional map* (FM) [39] is just a linear map $T \in \mathbb{R}^{K' \times K}$ sending functions $\mathbf{r}$ defined on $S$ to functions $\mathbf{r}' = T\mathbf{r}$ defined on $S'$. The space of such maps is very large, but we are interested particularly in two kinds. The first are point-to-point maps, which are analogous to permutation matrices $\Pi$, and thus express correspondences between the two shapes. The second kind are maps restricted to low-pass functions on the two surfaces. Assuming that functions can be written as $\mathbf{r} = U\hat{\mathbf{r}}$ and $\mathbf{r}' = U'\hat{\mathbf{r}}'$ for LBO bases $(U, A)$ and $(U', A')$, the functional map $\mathbf{r}' = T\mathbf{r}$ can be written in an equivalent manner as $\hat{\mathbf{r}}' = C\hat{\mathbf{r}}$ where $C = (U')^\top A'TU$ acts on the spectra of the functions. The advantage of the spectral representation is that the $M' \times M$ matrix $C$ is generally much smaller than the $K' \times K$ matrix $T$. Then, we can relate positional embeddings $E$ and $E'$ for the two shapes, which are smooth, as $\hat{E}' \approx C\hat{E}$.

When shapes $S$ and $S'$ are approximately isometric, we can resort to automatic methods to establish correspondences $\Pi$ (or $C$) between them. When $S$ and $S'$ are not, this is much harder. Instead, we

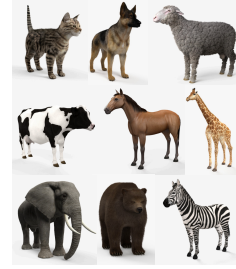

| category | LVIS # inst. | training set # inst. | # corresp. | coverage | | rage |
|---|---|---|---|---|---|---|
| dog | 2317 | 483 | 1424 | 21.5% | | 1% |
| cat | 2294 | 586 | 1720 | 23.9% | | 0% |
| bear | 776 | 98 | 289 | 4.8% | | 9% |
| sheep | 1142 | 257 | 765 | 13.3% | | 2% |
| cow | 1686 | 426 | 1267 | 19.7% | | 9% |
| horse | 2299 | 605 | 1783 | 25.8% | | 0% |
| zebra | 1999 | 665 | 1968 | 28.8% | | 6% |
| giraffe | 2235 | 651 | 1936 | 27.4% | | 2% |
| elephant | 2242 | 670 | 1992 | 28.1% | 200 592 | 10.0% |

Table 1: **Annotation statistics of the DensePose-LVIS dataset.** 'Coverage' is expressed as the number of vertices in a given class mesh with at least one corresponding ground truth annotation. The corresponding animal meshes are shown on the right (source: `hum3d.com`).

start from a small number of manual correspondences $(k_j, k'_j)$, $j = 1, \ldots, Q$ between surfaces and interpolate those using functional maps. To do this, we use a variant of the ZoomOut method [34] due to its simplicity. This amounts to alternating two steps: given a matrix $C$ of order $M_1 \times M_1$, we decode this as a set of point-to-point correspondences $(k_j, k'_j)$; then, given $(k_j, k'_j)$, we estimate a matrix $C$ of order $M_2 > M_1$, increasing the resolution of the match. This is done for a sequence $M_t = 12, 16, \ldots, 256$ until the desired resolution is achieved (see Appendix A.2 for details).

### 3.3 Cross-species DensePose with functional maps

We are now ready to describe how functional maps can be used to facilitate transferring and sharing a single DensePose predictor $\Phi$ between categories with different canonical shapes $S, S'$ and corresponding per-vertex embeddings $E, E'$.

To this end, assume that we have learned the pose regressor $\Phi$ on a source category $(S, E)$ (e.g. humans). We are free to apply $\Phi$ to an image $I'$ of a different category $(S', E')$ (e.g. chimps), but, while we might know $S'$, we do not know $E'$. However, if we assume that the regressor $\Phi$ can be shared among categories, then it is natural to also *share their positional embeddings* as well. Namely, we assume that $E'$ is approximately the same as $E$ up to a remapping of the embedding vectors from shape $S$ to shape $S'$. Based on the last section, we can thus write $\hat{E}' = C\hat{E}$, or, equivalently, $E' = TE$ where $T = U'CU^\top A$. With this, we can simply replace $E$ with $E'$ in eq. (2) to now regress the pose of the new category using the same regressor network $\Phi$. For training, we optimise the same cross entropy loss $\mathcal{L}(E, \Psi)$ in eq. (3), just combining images and annotations from the two object categories and swapping $E$ and $E'$ depending on the class of the input image.

The procedure above can be easily generalised to any number of categories $S^1, \ldots, S^{\mathcal{K}}$ with reference to the same source category $S$ and functional maps $C^1, \ldots, C^{\mathcal{K}}$. In our case, we select humans as source category as they contain the largest number of annotations.

## 4 Datasets

We rely on the DensePose-COCO dataset [17] for evaluation of the proposed method on the human category and comparison with the DensePose (IUV) training. For the multi-class setting, we make use of a recent DensePose-Chimps [44] test benchmark containing a small number of annotated correspondences for chimpanzees. We split the set of annotated instances of [44] into 500 training and 430 test samples containing 1354 and 1151 annotated correspondences respectively.

Additionally, we collect correspondence annotations on a set of 9 animal categories of the LVIS dataset [21]. Based on images from the COCO dataset [31], LVIS features significantly more accurate object masks. We refer to this data as DensePose-LVIS. The annotation statistics for the collected animal correspondences are given in Table 1. Note that compared to the original DensePose-COCO labelling effort that produced 5 million annotated points for the human category (96% coverage of the SMPL mesh), our annotations are three orders of magnitude smaller and only 18% of vertices of animal meshes, on average, have at least one ground truth annotation.

|  | architecture | AP | $\text{AP}_{50}$ | $\text{AP}_{75}$ | $\text{AP}_M$ | $\text{AP}_L$ | AR | $\text{AR}_{50}$ | $\text{AR}_{75}$ | $\text{AR}_M$ | $\text{AR}_L$ |
|---|---|---|---|---|---|---|---|---|---|---|---|
| IUV (baselines) | DP-RCNN (R50) [17] | 54.9 | 89.8 | 62.2 | 47.8 | 56.3 | 61.9 | 93.9 | 70.8 | 49.1 | 62.8 |
| | DP-RCNN (R101) [17] | 56.1 | 90.4 | 64.4 | 49.2 | 57.4 | 62.8 | 93.7 | 72.4 | 50.1 | 63.6 |
| | Parsing-RCNN [56] | 65 | 93 | 78 | 56 | 67 | – | – | – | – | – |
| | AMA-net [20] | 64.1 | 91.4 | 72.9 | 59.3 | 65.3 | 71.6 | 94.7 | 79.8 | 61.3 | 72.3 |
| IUV | DP-RCNN* (R50) | 65.3 | 92.5 | 77.1 | 58.6 | 66.6 | 71.1 | 95.3 | 82.0 | 60.1 | 71.9 |
| | DP-RCNN* (R101) | 66.4 | 92.9 | 77.9 | 60.6 | 67.5 | 71.9 | 95.5 | 82.6 | 62.1 | 72.6 |
| | DP-RCNN-DeepLab* (R50) | 66.8 | 92.8 | 79.7 | 60.7 | 68.0 | 72.1 | 95.8 | 82.9 | 62.2 | 72.4 |
| | DP-RCNN-DeepLab* (R101) | 67.7 | 93.5 | 79.7 | **62.6** | 69.1 | 73.6 | 96.5 | 84.7 | **64.2** | 74.2 |
| CSE | DP-RCNN* (R50) | 66.1 | 92.5 | 78.2 | 58.7 | 67.4 | 71.7 | 95.5 | 82.4 | 60.3 | 72.5 |
| | DP-RCNN* (R101) | 67.0 | 93.8 | 78.6 | 60.1 | 68.3 | 72.8 | 96.4 | 83.7 | 61.5 | 73.6 |
| | DP-RCNN-DeepLab* (R50) | 66.6 | 93.8 | 77.6 | 60.8 | 67.7 | 72.8 | 96.5 | 83.1 | 62.1 | 73.5 |
| | DP-RCNN-DeepLab* (R101) | **68.0** | **94.1** | **80.0** | 61.9 | **69.4** | **74.3** | **97.1** | **85.5** | 63.8 | **75.0** |

Table 2: **Performance on DensePose-COCO, with `IUV` (top) and `CSE` (bottom) training** (GPSm scores, `minival`). First block: published SOTA DensePose methods, second block: our optimized architectures + `IUV` training, third block: our optimized architectures + `CSE` training. All `CSE` models are trained with loss $\mathcal{L}_\sigma$ (eq. 4), LBO size $M = 256$, embedding size $D = 16$.

| # LBO basis, $M$ | loss $\mathcal{L}$ | loss $\mathcal{L}_\sigma$ |
|---|---|---|
| 32 | 62.9 | 63.2 |
| 64 | 64.1 | 63.9 |
| 128 | 65.2 | 65.4 |
| 256 | **65.6** | **66.1** |
| 512 | **65.6** | 65.9 |
| 1024 | 65.7 | 65.9 |

| # embedding, $D$ | loss $\mathcal{L}$ | loss $\mathcal{L}_\sigma$ |
|---|---|---|
| 2 | 38.3 | 46.4 |
| 4 | 60.0 | 64.7 |
| 8 | 60.2 | 65.6 |
| 16 | 65.4 | **66.1** |
| 32 | **65.6** | 66.0 |
| 64 | 65.1 | **66.1** |

| training data, % | training mode IUV | training mode loss $\mathcal{L}$ | training mode loss $\mathcal{L}_\sigma$ |
|---|---|---|---|
| 1 | 18.9 | 7.2 | 17.8 |
| 5 | 36.6 | 26.5 | 31.4 |
| 10 | 42.2 | 36.3 | 39.3 |
| 50 | 58.0 | 56.3 | 58.5 |
| 100 | 65.3 | 65.4 | 66.1 |

Table 3: **Hyperparameter search and performance in low data regimes** (AP, DensePose-COCO, `minival`): (left) LBO basis size, $M$ ($D = 16$), (center) embedding size, $D$ ($M = 256$), (right) comparison of `IUV` and `CSE` training in small data regimes. DP-RCNN* (R50) predictor.

# 5 Experiments

**Architectures.** Our networks are implemented in PyTorch within the Detectron2 [54] framework. The training is performed on 8 GPUs for 130k iterations on DensePose-COCO (standard `s1x` schedule [54]) and 5k iterations on DensePose-Chimps and DensePose-LVIS. The code, trained models and the dataset will be made publicly available to ensure reproducibility.

Prior to benchmarking the `CSE` setup, we carefully optimized all core architectures for dense pose estimation and introduced the following changes (similarly to [56, 20]): (1) single channel instance mask prediction as a replacement for the coarse segmentation of [17]; (2) optimized weights for $(i, u, v)$ components; (3) DeepLab head and Panoptic FPN (similarly to [43]). The experimental results are reported following the updated protocol based on GPSm scores [54]. More details on the network architectures and training hyperparameters are given in the supplementary material.

**Comparison of `CSE` vs `IUV` training.** The comparison of the state-of-the-art methods for dense pose estimation and our optimized architectures for both `IUV` and `CSE` training is provided in Table 2. The `CSE`-trained models perform better or on par with their `IUV`-trained counterparts, while producing a more compact representation ($D = 16$ vs $D = 75$) and requiring only simplified supervision (single vertex indices vs $(i, u, v)$ annotations).

**Influence of hyperparameters.** In Table 3 we investigate the `CSE` network sensitivity to the size of the LBO basis, $M$ (left), and the output embedding, $D$ (right), given training losses $\mathcal{L}$ or $\mathcal{L}_\sigma$. The value $M = 256$ represents the tradeoff between mapping's smoothness and its fidelity. It does not seem beneficial to increase the embedding size beyond $D = 16$, so we adapt this value for the rest of the experiments. The smoothed loss $\mathcal{L}_\sigma$ yields better performance in a low dimensional setting.

**Low data regime.** Prior to proceeding with the multi-class experiments on animal classes, we investigate changes in models performance as a function of the amount of ground truth annotations by training on subsets of the DensePose-COCO dataset. As shown in Table 3 on the right, $\mathcal{L}_\sigma$-based training scales down more gracefully and is significantly more robust than $\mathcal{L}$.

| model | $\mathcal{S}_{\text{chimp}}$ | $\mathcal{S}_{\text{smpl}}$ |
|---|---|---|
| human model | – | 2.0 |
| loss $\mathcal{L}$ | 8.5 | 3.3 |
| loss $\mathcal{L}_\sigma$ | 21.1 | 3.2 |
| + pretrain $\Phi$ | 36.7 | 34.5 |
| + align $E$ | **37.2** | **35.7** |

Table 4: **Performance on the DensePose-Chimps dataset with** CSE **training** (**AP**, GPSm scores, measured on both chimp and SMPL meshes wrt the GT mapping $\mathcal{S}_{\text{chimp}} \to \mathcal{S}_{\text{smpl}}$ from [43]).

| | model | cat | dog | bear | sheep | cow | horse | zebra | giraffe | elephant | mean |
|---|---|---|---|---|---|---|---|---|---|---|---|
| | single class + $\mathcal{L}$ | 5.5 | 4.7 | 1.8 | 0.9 | 2.8 | 4.5 | 12.5 | 11.4 | 19.4 | 7.05 |
| | single class + $\mathcal{L}_\sigma$ | 20.2 | 16.4 | 10.8 | 14.2 | 22.5 | 24.3 | 23.9 | 27.1 | 26.4 | 20.6 |
| joint training | multiclass + $\mathcal{L}_\sigma$ | 20.2 | 18.3 | 19.3 | 25.4 | 22.4 | 26.3 | 33.2 | 30.9 | 29.9 | 25.0 |
| | class agnostic + $\mathcal{L}$ | 8.7 | 6.6 | 4.1 | 8.2 | 6.4 | 7.3 | 19.2 | 15.5 | 9.2 | 9.5 |
| | class agnostic + $\mathcal{L}_\sigma$ | 20.5 | 18.3 | 20.1 | 25.9 | 24.5 | 25.7 | 34.5 | 30.5 | 27.1 | 25.2 |
| | + pretrain $\Phi$ | 28.0 | 28.3 | 22.0 | 31.8 | **36.5** | 32.7 | 43.1 | **41.2** | 34.9 | 33.1 |
| | + align $E$ | **30.9** | **29.4** | **25.1** | **35.3** | **36.5** | 34.3 | **46.0** | 38.3 | **39.6** | **35.0** |

Table 5: **Performance on the DensePose-LVIS dataset with** CSE **training** (**AP**, GPSm scores).

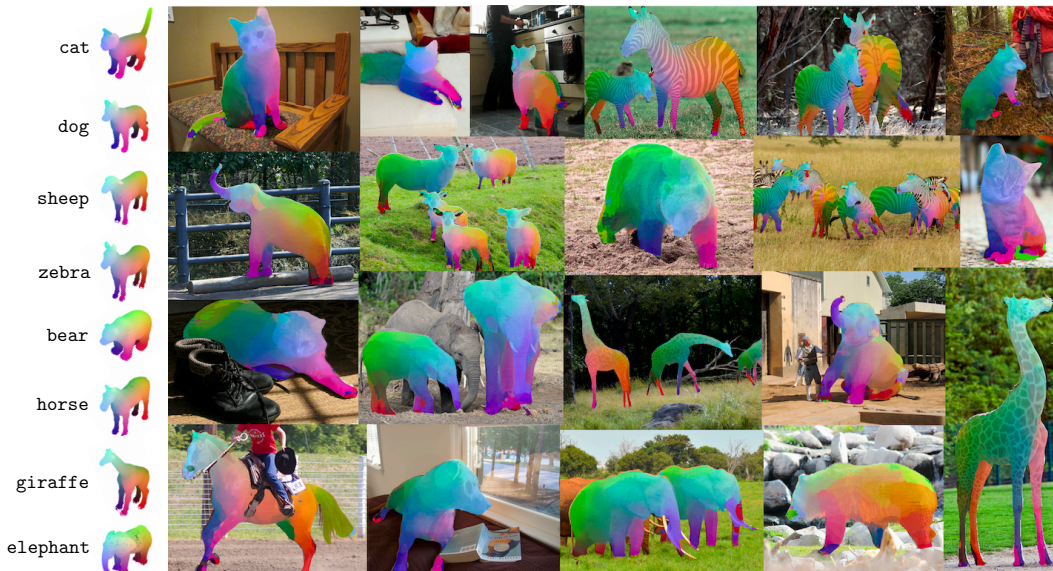

Figure 4: **Qualitative results on the DensePose-LVIS dataset** (single predictor for all classes).

**Multi-surface training.** The results on DensePose-Chimps and DensePose-LVIS datasets are reported in Tables 4 and 5 (DP-RCNN* (R50), $M = 256$, $D = 16$). In both cases, training from scratch results in poor performance, especially in a single class setting. Initializing the predictor $\Phi$ with the human class trained weights together with the alignment of mesh-specific vertex embeddings (as described in 3.3) gives a significant boost. Interestingly, class agnostic training by mapping all class embeddings to the shared space turns out to be more effective than having a separate set of output planes for each category (the latter is denoted as *multiclass* in Table 5). Quantitative results produced by the best predictor are shown in Figure 4.

**Conclusion.** In this work, we have made an important step towards designing universal networks for learning dense correspondences within and across different object categories (animals). We have demonstrated that training joint predictors in the image space with simultaneous alignment of canonical surfaces in 3D results in an efficient transfer of knowledge between different classes even when the amount of ground truth annotations is severely limited.

## Broader impact

In our paper, we help improve the ability of machines to understand the pose of articulated objects such as humans in images. In particular, we make the process of learning new object categories much more efficient.

An application of our method is the observation of the human body. This may come with some concerns on possible negative uses of the technology. However, we should note that our approach cannot be considered biometrics, because from pose alone, even if dense, it is not possible to ascertain the identity of an individual (in particular, we do not perform 3D reconstruction, nor we reconstruct facial features). This mitigates the potential risk when our method is applied to humans.

We believe that our work has significant opportunities for a positive impact by opening up the possibility that machines could ultimately understand the pose of thousands of animal classes. In addition to numerous applications in VR, AR, marketing and the like, such a technology can benefit animal-human-machine interaction (e.g. in aid of the visually impaired), can be used to better safeguard animals on the Internet (e.g. by detecting animal abuse), and, perhaps most importantly, can allow conservationists and other researchers to observe animals in the wild at an unprecedented scale, automatically analysing their motion and activities, and thus collecting information on their number, state of health, and other statistics. Thus, while we acknowledge that this technology may find negative uses (as almost any technology does), we believe that the positives far outweigh them.

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
