[Supplementary Material]

# Continuous Surface Embeddings

## 1   Appendix A

### 1.1   Discrete Laplace-Beltrami operator

**Laplace-Beltrami operator** $(W, A)$**.**   In order to construct the discrete Laplace-Beltrami Operator (LBO), and with reference to the notation introduced in the main manuscript, we assume that the points $X_k$ are the vertices of a *simplicial mesh*, i.e. the union $\bar{S} = \cup_{f \in F} f$ of a finite set $F$ of triangular faces $f$, approximating the 3D surface $S$. With slight abuse of notation, we denote each face $f = (X_1, X_2, X_3)$ as a triplet of vertices oriented in clockwise order with respect to the normal $N_f$ of the face. If we assume that the function $r$ is continuous and linear within each face, then the samples  fully specify the function. Let $b_1 = X_3 - X_2$, $b_2 = X_1 - X_3$ and $b_3 = X_2 - X_1$, be the edge vectors opposite to each vertex of the triangle $f$ and let $A_f$ be its area. The gradient of $r$, which is constant on each face $f$, is given by:

$$(\nabla r)_f = \frac{1}{2A_f} \sum_{i=1}^{3} (N_f \times b_i) r_{f_i} = G_f f, \quad f = \begin{bmatrix} r_{f_1} \\ r_{f_2} \\ r_{f_3} \end{bmatrix} \quad A_f = \frac{1}{2} |b_1 \times b_2|, \quad N_f = \frac{1}{2A_f} (b_1 \times b_2). \quad (1)$$

The *Dirichlet energy* of the function $r$ is the integral of the squared gradient norm: $\int_{\bar{S}} \|\nabla r\|^2 \, dS = \sum_{f \in F} A_f \|(\nabla r)_f\|^2 = \sum_{f \in F} {}^{\top} W_f f$, where $A_f$ is the area of face $f$ and $W_f = A_f G_f^{\top} G_f$. By summing $L_f$ over the faces, we obtain the overall LBO operator $W \in \mathbb{R}^{K \times K}$ mapping  to the total Dirichlet energy ${}^{\top} W$ (since this energy is non-negative, $W$ is positive semi-definite). We also need the diagonal matrix $A$ of *lumped areas*, with $A_{kk}$ being a third of the total areas of the triangles incident on vertex $X_k$. [1]

**Gradient operator** $G_f$**.**   We can verify the expression eq. (1) for the gradient as follows.  The gradient dotted with an edge vector $b_i$ must give the function change along that edge. For example, for edge $b_1$ we have:

$$\langle b_1, (\nabla r)_f \rangle = \sum_{i=1}^{3} \frac{\langle b_1, N_f \times b_i \rangle}{2A_f} r_{f_i} = \frac{\langle N_f, b_2 \times b_1 \rangle}{2A_f} r_{f_2} + \frac{\langle N_f, b_3 \times b_1 \rangle}{2A_f} r_{f_3} = r_{f_3} - r_{f_2}.$$

We can write matrix $G_f$ in eq. (1) much more compactly as:

$$G_f = \frac{1}{2A_f} \hat{N}_f B_f, \quad B_f = B_f \hat{\mathbf{1}}^{\top}, \quad V_f = \begin{bmatrix} X_1 & X_2 & X_3 \end{bmatrix}, \quad \hat{a} = \begin{bmatrix} 0 & -a_3 & a_2 \\ a_3 & 0 & -a_1 \\ -a_2 & a_1 & 0 \end{bmatrix},$$

Here $\hat{\cdot}$ is the hat operator, such that $a \times b = \hat{a}b$, and and $\mathbf{1} = (1, 1, 1)$, so that $B_f = \begin{bmatrix} b_1 & b_2 & b_3 \end{bmatrix}$. By summing $G_f$ over the faces $f$, we obtain an discrete operator $G \in \mathbb{R}^{3|F| \times K}$ mapping the function to the gradient in each face.

**Cotangent weight matrix $W$.** Given the expression for $G_f$, we can find a compact expression fo $W_f$ in the LBO:

$$W_f = A_f G_f^\top G_f = \frac{1}{4A_f} B_f^\top B_f.$$

Note that $B_f^\top \hat{N}_f^\top \hat{N}_f B_f = B_f^\top B_f$ because $b_i \perp N_f$ and thus:

$$\langle N_f \times b_i, N_f \times b_j \rangle = \langle N_f, N_f \rangle \langle b_i, b_j \rangle - \langle N_f, b_j \rangle \langle N_f, b_i \rangle = \langle b_i, b_j \rangle.$$

$W_f$ matches the usual *cotangent discretization* of the Laplace-Beltrami operator: $B_f$ contains dot products of edges, $2A_f$ the norm of their cross products, and the ratio of these two are cotangents.

**Divergence operator $D$.** Finally, we sometimes require a divergence operator. For this, let $X_f \in \mathbb{R}^3$ a vector defined on each face. In order to compute the divergence at a vertex $X_1$, we find the contour integral of $X_f$ along the boundary of the triangle fan centered at $X_1$. Thus let $f = (X_1, X_2, X_3)$ be a face belonging to this fan and $b_1, b_2, b_3$ be the corresponding edge vectors as before. The contribution of this triangle to the contour integral around $X_1$ is:

$$D_v X_f = |b_2| \cdot \langle n, X_f \rangle, \quad n|b_2| = b_2 \times N_f = \frac{1}{2A_f} b_2 \times (b_3 \times b_1) = b_3 \frac{\langle b_2, b_1 \rangle}{2A_f} - b_1 \frac{\langle b_2, b_3 \rangle}{2A_f}$$

## 1.2 Spectra interpolation of correspondences (ZoomOut)

Assume that we have complete correspondences for the mesh $S'$, in the sense that $k'_j = j$, $j = 1, \ldots, K'$. We can encode those as a permutation matrix $\Pi$ such that $\Pi_{ij} = \delta_{k_i = j}$, mapping functions on $S$ to function $' = \Pi$ on $S'$ (this is analogous to backward warping). This can be rewritten in 'Fourier' space as $' = U'' = U'C = \Pi = \Pi U$, which gives us the constraint [1] $U'C = \Pi U$. We can use this equation to find $C$ given $\Pi$, or to find $\Pi$ given $C$. Finding $\Pi$ is done in a greedy manner, searching, for each row of $U'C$, the best matching row in $U$ (in $L^2$ distance). Finding $C$ is done by minimizing $\|U'C - \Pi U\|_{A'}^2$, which results in $C = (U')^\top A' \Pi U$.

In practice, we found it beneficial to add three more standard constraints when resolving for $C$. First, let $\Gamma \in \{0, 1\}^{K \times K}$ be the symmetry matrix mapping each vertex of mesh $S$ to its symmetric counterpart (this is trivially determined for our canonical models), and let $\Gamma'$ be the same for $S'$. Then a correct correspondence $\Pi$ between meshes must preserve symmetry, in the sense that $\Gamma'\Pi = \Pi\Gamma$; this constraint can be rewritten in Fourier space as $\hat{\Gamma}'C = C\hat{\Gamma}$, where $\hat{\Gamma} = U\Gamma U^\dagger$. For isometric meshes, the exact same reasoning applies to the LBO $L = A^{-1}W$ because the LBO is an intrinsic property of the surface (i.e. invariant to isometry). Our meshes are not isometric, but, after resizing them to have the same total area, we can use the constraint $L'\Pi \approx \Pi L$ in a soft manner for regularization; it is easy to show that this reduces to $\Lambda'C \approx C\Lambda$ where $\Lambda$ is the matrix of eigenvalues of the LBO. In practices, this encourages $C$ to be roughly diagonal. Finally, we use the method just described twice, to estimate jointly a mapping $C$ from mesh $S$ to $S'$, and another $C'$ going in the other direction, and enforce $CC' \approx I$ (cycle consistency).

# 2 Appendix B

## 2.1 Annotation process

We are following an annotation protocol similar to the one described in the original DensePose work [2]. We start with instance mask annotations provided in the LVIS dataset and crop images around each instance. We only annotate instances with bounding boxes larger than 75 pixels. We do not collect annotations for body segmentation: instead, the points are sampled from the whole foreground region represented by the object mask. The annotators are then shown randomly sampled points displayed on the image and are asked to click on corresponding points in multiple views rendered from a 3D model representing the given species. Each worker is asked to annotate 3 points on a single object instance. The points on the rendered views are mapped directly to vertex indices of the corresponding model. Each mesh is normalised to have approximately 5k vertices.

## 2.2 Implementation details

Compared to the original DensePose [3] models, we introduced the following changes:

- single channel mask supervision as a replacement of the 15-way segmentation;
- RoI pooling size for the DensePose task is set to $28 \times 28$;
- a decoder module based on Panoptic FPN, as implemented in [4];
- for the DeepLab models, the head architecture corresponds to [4];
- IUV training, weights of individual loss terms: $w_{mask} = 5.0$ (body mask), $w_i = 1.0$ (point body indices), $w_{uv} = 0.01$ (*uv* coordinates);
- CSE training, weight on the embedding loss term: $w_e = 0.6$.

For the DensePose-COCO dataset, all models are trained with the standard s1x schedule [5] for 130k iterations. On the LVIS and DensePose-Chimps datasets, the models are trained for 5k iterations with the learning rate drop by the factor of 10 after 4000k and 4500k iterations.

For the evaluation purposes all 3D meshes are normalised in size to have the same geodesic distance between the pair of most distant points as the SMPL model ($P_{\text{dist.,max}} = 2.5$). For the animal classes, we do not employ part specific normalisation coefficients, as done in the updated DensePose evaluation protocol.

The code, the pretrained models and the annotations for the LVIS dataset will be publicly released.

## Footnotes

[1] This is a required normalization factor to account for the different areas of the triangles w.r.t. the continuous surface being approximated.