[Reviews · NeurIPS 2020]

Review 1

Summary and Contributions: The paper performs some alternations to the DensePose method, to make it require less manual annotations, and perform slightly better. The method basically offers to learn the flattening (or charting) requirement imposed by DesnePose, thus making it more scalable and flexible. The main idea is based on functional maps. As an added bonus, the method can use one predictor for several classes.

Strengths: The problem addressed by the paper is an important and timely one, as many other works try to tackle the problems associated with learning over 3D. In my opinion, the biggest strength the method has to offer is the fact that it yields a single predictor for several classes. Further more, the method offers a slight improvement over DensePose even on a single class.

Weaknesses: The largest weakness of the paper is its contribution. The paper describes a not-so-easy-to-implement addition to the DensePose system, which yields very little to no improvement in the results. The paper does try in any way to justify the choice of the addition, or claim that this approach might be better than other options. For example, could one automate the DensePose system by traditional co-segmentation and traditional parameterization approaches as a preprocess? Would it be possible, using the minimal annotation described in this paper, to compute some other translation between charts in DensePose in order to give it more flexibility and some cross-class scalability (maybe even using functional maps)? After reading the rebuttal, some of the questions I had about design choices have been put to rest, but I'm still, unfortunately, not convinced this paper is impactful enough for a NeurIPS publication, so my score did not change.

Correctness: yes

Clarity: yes, although some more implementation and training details could be nice, maybe in the supplementary material. This is, of course, not a pressing issue should the code be released.

Relation to Prior Work: The paper claims that DensePose is the state-of-the-art, and compares only to it. I am not an expert in this exact problem, but a quick search yielded many papers that seems relevant, that are newer, and some even also use functional maps. The authors should either compare to these works, or distinguish themselves very clearly from them. Here are a few examples: Unsupervised learning of dense shape correspondence, 2019 MINA: Convex Mixed-Integer Programming for Non-Rigid Shape Alignment, 2020 PRNet: Self-supervised learning for partial-to-partial registration, 2019

Reproducibility: Yes

Additional Feedback: In its current form, the paper, has very little scientific justification and comparisons. That said, the paper seems like it has an interesting idea at its core. For these reasons, I am recommending against its acceptance, but only marginally.


Review 2

Summary and Contributions: The paper addresses the problem of finding (dense) correspondences between 2D images and corresponding 3D object surfaces. This is achieved by learning an embedding function which maps points of the surface (vertices in the discrete variant) to "embedding descriptors", and a deep NN which maps image pixels to embedding descriptors. The loss function for training effectively uses a softmax-like probabilistic measure of goodness of fit between the two. Further, LBO bases are used to reduce the number of learnable parameters.

Strengths: (1) Simple and elegant solution. Theoretical grounding is sound. (2) Works well with substantially smaller amount of annotations than with other models. (3) Very favourably compares with state-of-the-art models in terms of performance. (4) Addresses the cumbersome problem of charting in DensePose model. (5) Results are transferable to unseen object categories. (6) Empirical evaluation is extensive.

Weaknesses: I do not see major limitations of the work.

Correctness: The claims and methods appear correct to me. The empirical methodology is sound and experiments are extensive.

Clarity: The paper is very well written, with good structure and attention to detail.

Relation to Prior Work: The paper gives a sound overview of the state-of-the-art, and especially clearly addresses the differences from the competing DensePose model.

Reproducibility: Yes

Additional Feedback:


Review 3

Summary and Contributions: The paper proposed to extend the existing DensePose effort of 2D-alone prediction of dense correspondence to 2D-3D joint prediction where the 3D part is implemented by 3D shape geometry analysis. The proposed approach is intuitive and well-motivated. The author contributed to a new DensePose-LVIS dataset with new correspondence annotations. Extensive experiments are performed and show leading results on the DensePose-COCO dataset.

Strengths: The contribution is novel and significant enough: associating 3D shape analysis to 2D coordinate maps is an intuitive yet not well-explored idea. The claims are sound and are relevant to the NeurIPS community. The description of the approach is clear. Using spectral analysis for better shape understanding is an elegant idea. Experiment results are thorough and convincing. It also shows that the model can generalize to new categories from new meshes.

Weaknesses: The author could also include a model run time analysis to show how introducing 3D geometry analysis part affects the total run time.

Correctness: Yes.

Clarity: Yes.

Relation to Prior Work: Yes.

Reproducibility: Yes

Additional Feedback:


Review 4

Summary and Contributions: This paper proposes a method for dense correspondence prediction (from 2D image) for deformable objects, such as human and animals. Based on my understanding, the work is based on state-of-the-art method DensePose, but changes the following aspect: 1. Redefines the representation for correspondence descriptors from IUV (body part index, and uv coordinates in corresponding altas island) to a continuous descriptors (that can be regarded as 2-manifold in D dimension) 2. A transferring mechanism that potentially allows prediction on multiple categories. The DensePose-style network predicts a “universal” descriptor map, which pre-processed transferring functions are applied to transfer it to desired category. The transfer is done by existing method functional map (with truncated Laplacian-Beltrami basis) and solved by ZoomOut. The work shows qualitative and quantitative comparison to baseline methods (based on DensePose) and presents improved performances. The work also shows the transferred correspondence analysis on several mammal animals categories.

Strengths: I think this paper explores an interesting and potential useful problem setting. I agree that the original IUV formulation as in original DensePose was a bit problematic. It would be better to define the continuous descriptors due to the nature of the data (human body surface can be regarded as 2-manifold instead of connected body parts). The proposed system presents some novelty and empirical analysis shows some improvement. The proposed dataset could be helpful in research in this area.

Weaknesses: Since I found the exposition of this paper is not very clear in some parts, I would ask questions on the settings / designs, and then raise concerns. Please confirm/explain if anything. 1. Are the transferring functions between categories (e.g. human->cat) precomputed on two specific template mesh of human and cat, or two sets of mesh instances? If it’s from two specific template meshes, then I doubt the transformation estimated from functional maps would be general enough for transforming different types of cats. Further, I recall the LB basis doesn’t stay static (in corresponding location on surface) if the object mesh is deforming. Therefore the proposed method to get transformation would also not fit for cats in different poses (rigid and non-rigid). In this sense, the design of the algorithm in this part is already not suitable for the problem that aims to cover deformable objects in various appearances and poses. 2. The transfer function from “universal” descriptor to target descriptor is designed to be linear. Do you think the linear transfer is sufficient? Since the appearance of different animals are quite different, I doubt so. (regardless, this would be an interesting experiment) 3. Would you show the solved correspondence transfer in visual results? The difficult parts would be those parts for which humans are not significant (tails, ear, nose->elephant trunk). Those parts might be definable “anatomically” (humans have a tail bone corresponding to long tails of many animals), but the mapping will result in quite extreme deformation in correspondence (not isometric). 4. On results: I found the improvement from baseline (e.g. in table 2) is quite marginal. Table 2: the improvement of best cases of CSE to corresponding setting of baseline are mostly <1.0 in all metrics. The visual results also presents potential mismatches: Figure 4: Row 1 example 1, the feet of cats are mistaken. Similar issue in Row 1 example 6, Row 2 example 5, giraffe feet in Row 3 example 3 The elephant trunk looks very similar to the color mapping of feet. I see artifacts that look “blocky”, such as row 2 example 3, row 4 example 4. 5. Improve visualization: might be good add the visualization that you would randomly sample some point on source image / scan and draw a line to the matched point by the proposed method. One more (slightly ambitious) visualization is to use the predicted descriptors to drive the template mesh (similar to the one in DensePose video results). This is actually a good application case, in that you could track animal motion from a single view image / video. 6. Improve writing: To me I feel the explanation on Laplacian-Beltrami operators and functional maps can be more concise or moved to supplement materials. Would be good to reorganize the procedure of compute the transfer and add more results.

Correctness: Please see the weakness session.

Clarity: Please see the weakness session.

Relation to Prior Work: This work reminds me of this paper Learning to Segment Every Thing, CVPR 18, Hu et al. Where they learn a non-linear transform function between the detection weights to segmentation weights for each category. To me due to similarity between detection and segmentation, the transfer between the two should be simple, yet they learn a non-linear MLP to make it work. Therefore I doubt the transfer appearance->correspondence descriptor among categories would work with merely a linear function. I understand the settings are quite different: Hu et al transfers parameter weights while here the authors transfers the descriptor (outputs). Just adding some context here. I believe there could be more discussion of this work to those in the few-shot learning area, as the setting of this problem is quite similar.

Reproducibility: No

Additional Feedback: ** After rebuttal ** Thank you for the very detailed rebuttal. After reading the rebuttal, I still hold some concerns the design of the method. The main concern is whether or not the inter-class transfer map can generalize for animals in various poses and shapes is questionable. As I raised in my original review, the LB basis is not "static" if mesh is not isometrically deformed. One example is shown in Michael Bronstein's presentation at https://youtu.be/JtBWzI3OaxI?t=598, at 9:58. Probably for similar reasons, I see more artifacts in the visualization of the dense pose prediction (Fig 4), when the animal is heavily deformed and not in the "standard pose" (as in the canonical meshes in Fig 4 left column). Plus, for animal within the same category but different shapes (different breeds or sizes), the LB basis would be different. Therefore the inter-class transfer map precomputed on *1* specific mesh per category would be problematic. The problem setting aim to predict dense pose descriptors for deformable objects, however the design of the method seems not able to support for deformable objects in various shape and pose. Further the results seem to fit the issues: artifacts for non-canonical poses. The authors gave confirmations on the linear map; that is good. But regardless of this, the issue of LB basis and estimation on 1 particular example persists. However considering this is a brand new problem and the authors gave a reasonable solution (although with some flaws), I would raise the rating to 5: marginally below threshold. I would further encourage the authors to address the weakness by improving the results and discussing the potential weaknesses. More specifically, I would suggest to evaluate matching results with improved visualizations on inputs with animals in various shape and pose. I do see the improve visualization in the rebuttal (that's good), but they all seems very close to the canonical pose. This might fit my speculation that this method would work only on animals in canonical shape. Previous results (e.g. Fig 4) seems to have problem for non-canonical poses. If the results can provide clear improvement over the baseline, then I would think it's an acceptance. Apart from the dataset, the following paper seems to provide some good test data (with correspondence) for evaluation: Zuffi et al., Lions and Tigers and Bears: Capturing Non-Rigid, 3D, Articulated Shape from Images, CVPR 19. Further, this might be a bit ambitious, but visualizing predicted correspondence in 3D would be helpful. Using the descriptors to drive a template mesh, and if the induced motion is reasonable, this would help the paper a lot.

[Author Response · NeurIPS 2020]

**Reviewer 1** [does not] justify the choice of the addition There are two motivations for our continuous embeddings (see also lines 22 to 48). The first is to **simplify Dense Pose**. Charts in Dense Pose are a legacy from prior models such as DenseReg, which used them to improve inference accuracy *despite* the added complexity and issues such as the arbitrary seams between the charts. Our continuos embeddings remove charts with the same or even slightly better accuracy. The resulting model is simpler (fewer lines of code). Furthermore, the mesh pre-processing is not manual as in Dense Pose, but automatic, and can be obtained by using an off-the-shelf LBO implementation. The second motivation is to make it much easier to build **unified models of multiple objects** (lines 31 to 48). The original Dense Pose considered a single class (humans), and recent papers such as [44] had to do quite a bit of manual work just to tackle two classes (humans and chimps). Our method requires much less manual work because the continuous embeddings support functional maps, which are one of the most straightforward ways of modelling correspondences between 3D shapes (as they reduce the problem to simple linear algebra operations). **Not-so-easy to implement, automated co-segmentation** These are valid alternatives, but, respectfully, we do not see how these would be simpler than our approach. Our method makes implementing Dense Pose simpler and removes the need for segmentation charts, manually or automatically. Furthermore, note that most of the conceptual complexity is limited to relating different shapes via functional maps, which is not needed for modelling a single category. **Unsupervised learning of dense correspondences [22], other works.** These methods tackle the problem of 3D mesh alignment, not of detecting pose in 2D images as we do. In our setting, they could be used as a replacement component for ZoomOut (line 184) to establish the initial 3D mesh alignment (as a matter of fact, we did test [22] as a ZoomOut replacement, but found results to be unsatisfactory when aligning only a handful of meshes).

**Reviewer 3.** **Runtime analysis.** At test time our model is typically slightly faster than the standard Dense Pose network because the overall output dimensionality of the network is usually lower ($D$ vs number of charts by 3 in Dense Pose).

**Reviewer 4.** **(1) Only a single mesh per category** Dense Pose uses a single mesh for all humans, so one mesh per animal class is likely to be sufficient in most cases. Note that each mesh provides a canonical representation of dense correspondences, and as such it does not need to be geometrically faithful to any particular object instance. Instead, the mesh should be close enough to the object to allow human annotators to mark correspondences and to provide a weak geometric prior on the possible correspondences between different animals. **(1) LB doesn't stay static.** True, the LBO is invariant only to isometric deformations. Nevertheless, it is still heavily used in computational geometry even for meshes that are non-isometrically related, even in presence of major topological changes (see for example Functional map networks for analyzing and exploring large shape collections. Huang, Wang, Guibas. ACM Trans. Graph., 33(4):36:1–36:11, 2014.) This is because: (1) *part* of the LBO structure is approximately preserved even in these cases and (2) LBO is still useful as a generic 'smoothness' prior on the functional/correspondence maps even when it is not invariant. See also lines 181 to 188. **(2) transfer function…linear…sufficient?** Yes, they are sufficient. Note that these are *functional maps*, namely linear maps acting on the space of *functions* $S \to \mathbb{R}^d$, not merely on their *range* $\mathbb{R}^d$. In particular, functional maps include all warps of functions defined on a shape $S$ (in particular the embedding function $e_X$) to functions defined on a shape $S'$ (in our case $e'_{X'}$). Since we assume that the embeddings of two shapes are related by a warp, this relation is expressible as a (linear) functional map. Intuitively, in the discrete case, functional maps include all *permutations* matrices. Naturally, in a practical implementation one only considers a finite-dimensional subspace of all possible functional maps; this means that only sufficiently smooth warps can be represented, which, rather than being a problem, is good for regularization. If anything, we require additional constraints for the functional map to *only* express a warp (see also lines 483 – 494 in the appendix). **(3) extreme deformations.** We do agree that this aspect can be improved in the future. Nevertheless, 'Functional map networks for analyzing and exploring large shape collections' (see above) did show that even large topological changes can be handled by functional maps as these can collapse parts that are not in correspondence.

zebra → zebra        zebra → zebra

cat → zebra        elephant → zebra

**(4) baseline improvement marginal…mismatches…** This is the first paper able to extend Dense Pose to several classes using a single canonical space. Even for our baseline model, the new basic machinery (continuos embeddings, LBO) is still required. On top of this contribution, the prior arising form 3D mesh alignment improves result further. Finally, it is true that there are some mismatches, but we believe our results constitute overall a significant positive delta in this space. **(5) Visualization, template mesh** Thank you for the suggestion. Some preliminary examples using a higher-frequency procedural texture are given in the figure (we are working on implementing the use of arbitrary handcrafted textures). **(6) 'Learning to Segment Every Thing, CVPR 18': MPL vs linear function** See point (2): using a linear function*al* instead of an MLP is not really a limitation as these already include all useful warps (permutations, correspondences) of the embedding vectors between meshes.

[Meta-Review · NeurIPS 2020]

Two reviewers see value in the paper, while two are somewhat swayed by the rebuttal but find that the paper's overall contribution is not quite at the (high) bar that is set by NeurIPS. One point of widespread disagreement is about whether the method is "simple". The rebuttal clarifies that "simple" should mean fewer lines of code. However, there is no value in fewer lines of code per se -- the real value is that the new idea has theoretical interest or practical merit. On the former, there is some agreement "the paper seems like it has an interesting idea at its core", " explores an interesting and potential useful problem setting", but there is also general agreement that the improvement is relatively small, that artefacts still exist, and that the problem domain, although interesting, is relevant only to a subset of the NeurIPS audience. In that sense, it is reasonable for the reviewers to require more comprehensive explorations. On one point: the issue of linear function*al* vs linear function was understood by the reviewers, who also understand that a low-finite-dimensional representation of the functional might indeed not be as rich as an MLP. The rebuttal somewhat sidesteps this point, but it should be addressed much more openly in any final version of this paper.